# Uremic Toxins and Vascular Calcification–Missing the Forest for All the Trees

**DOI:** 10.3390/toxins12100624

**Published:** 2020-09-29

**Authors:** Nikolas Rapp, Pieter Evenepoel, Peter Stenvinkel, Leon Schurgers

**Affiliations:** 1Department of Biochemistry, Cardiovascular Research Institute Maastricht (CARIM), Maastricht University, 6229 ER Maastricht, The Netherlands; n.rapp@maastrichtuniversity.nl; 2Laboratory of Nephrology, KU Leuven Department of Microbiology and Immunology, University Hospitals Leuven, 3000 Leuven, Belgium; pieter.evenepoel@uzleuven.be; 3Karolinska Institute, Department of Clinical Science, Intervention and Technology, Division of Renal Medicine, 141 86 Stockholm, Sweden; peter.stenvinkel@ki.se

**Keywords:** uremic toxins, uremia, chronic kidney disease, cardiovascular disease, vascular smooth muscle cells, vascular calcification, middle molecules, protein bound uremic solutes, water-soluble uremic solutes

## Abstract

The cardiorenal syndrome relates to the detrimental interplay between the vascular system and the kidney. The uremic milieu induced by reduced kidney function alters the phenotype of vascular smooth muscle cells (VSMC) and promotes vascular calcification, a condition which is strongly linked to cardiovascular morbidity and mortality. Biological mechanisms involved include generation of reactive oxygen species, inflammation and accelerated senescence. A better understanding of the vasotoxic effects of uremic retention molecules may reveal novel avenues to reduce vascular calcification in CKD. The present review aims to present a state of the art on the role of uremic toxins in pathogenesis of vascular calcification. Evidence, so far, is fragmentary and limited with only a few uremic toxins being investigated, often by a single group of investigators. Experimental heterogeneity furthermore hampers comparison. There is a clear need for a concerted action harmonizing and standardizing experimental protocols and combining efforts of basic and clinical researchers to solve the complex puzzle of uremic vascular calcification.

## 1. Relevance and Objective

Chronic kidney disease (CKD) is recognized as a major non-communicable disease of growing epidemic dimension worldwide. The prevalence of CKD has increased by 19.6% between 2005 and 2015 [1] and reaches an overall global level of 11–13% [2]. Up to 70% of the healthcare cost related to CKD are due to hospitalization [3], the majority of which is accounted for by cardiovascular events [4]. The risk for cardiovascular disease (CVD) related hospitalization or death has been shown to increase with progression of CKD [5]. Dialysis increases cardiovascular mortality by up to 20-fold [6]. The reciprocal detrimental effects of the cardiovascular (CV) system and the kidneys on each other has been termed cardiorenal syndrome (CRS). CRS can be classified in five types according to duration and aetiology, namely: acute CRS, chronic CRS, acute renocardiac syndrome, chronic renocardiac syndrome and secondary CRS. CRS type 4, the chronic renocardiac syndrome, describes CKD leading to CVD [7]. While traditional Framingham risk factors like dyslipidaemia and diabetes contribute to CVD, they cannot fully explain the excessive mortality observed in patients with CRS type 4. Non-traditional risk factors including inflammation, oxidative stress or abnormal calcium-phosphate metabolism, have been found to account, at least partly, for the excessively high cardiovascular morbidity and mortality in these patients [8]. Non-traditional risk factors become more important along the progression of renal failure. Importantly, CKD patients are likely to die of CVD before even reaching end stage renal disease (ESRD) [9]. Prevalent CV phenotypes in CKD include atherosclerosis, cardiac arrythmia and vascular calcification (VC) [10]. VC affects up to 60% of CKD patients and is even more prevalent in dialysis patients [11]. Furthermore, it has been independently associated with CV morbidity and mortality [11,12]. Impaired renal function leads to retention of numerous compounds (referred to as uremic retention molecules, URMs). Information on the impact of URMs on vascular (patho) biology, in general, is limited and fragmentary, and pathophysiological mechanisms remain largely obscure. In this review, we present a state of the art on the effects of uremic retention molecules on vascular smooth muscle cells (VSMC) with relevance for early vascular ageing (EVA) and VC related CVD. Early vascular ageing-a modifiable, not passive entropic process [13]-is an evolving construct that has been growing around accumulating evidence surrounding arterial stiffness as an intermediate end-point and independent predictor of CVD. CKD-associated EVA is characterized by a loss of plasticity and/or resilience to adaptions against the changing internal uremic environment, which results in a marked discrepancy between chronological and biological vascular age [14]. In depth analysis of translational cohort studies investigating extreme EVA, such as CKD, provides valuable insights in factors that drive vascular ageing, and which may be also present in the older general population but take considerable longer time to evolve.

## 2. Vascular Calcification

VC as consequence of EVA can occur at two distinct sites: in the intimal layer of vessels where it is associated with atherosclerosis and in the medial layer, where it is also referred to as Mönckeberg sclerosis. Medial calcification is linked to non-traditional risk factors and is a common feature in CKD, which often co-exist with intimal calcification [15,16]. Arterial calcium depositions have long been perceived as a passive process strongly associated with aging. This common opinion shifted as more recent research showed that VC is an actively regulated process involving a delicate balance between calcification activators and inhibitors, present both in the vessel wall and in the circulation [17]. The medial layer of larger arteries is predominantly composed of VSMC, a differentiated cell type expressing a set of characteristic proteins (e.g., smooth muscle myosin heavy chain, sm22 alpha and myocardin). Contractile VSMC are significantly involved in maintaining vascular tone and structural integrity of the vessel wall. However, VSMC are not terminally differentiated and retain a high degree of phenotypic plasticity. Mechanical and/or chemical stress promote a phenotypic transition of the VSMC into a synthetic state, characterized by an increased ability to synthesize extracellular matrix components and accompanied by increased capacity for migration and proliferation [18]. Dedifferentiated VSMC have a variety of cell-fates, including macrophage-like, myofibroblast-like and osteo-chondrogenic-like phenotype [19]. Osteo-chondrogenic VSMC are characterized by a loss of contractility marker and an increase in osteo-chondrogenic marker proteins (e.g., Runt-related transcription factor 2 [Runx2], alkaline phosphatase [ALP], SRY-Box Transcription Factor 9 [Sox9], bone morphogenetic protein 2 [BMP2]). It has been reported that VSMC calcification is related to cellular senescence. Ref. [14] Furthermore the change in phenotype is accompanied by an increase in release of calcifying extracellular vesicles, elastin degradation and creation of a calcification-prone matrix [20]. Osteo-chondrogenic VSMC support the growth of hydroxyapatite (HA) crystals in the vessel wall, which defines calcification. Osteo-chondrogenesis and VC are driven by stressors including high phosphate, high calcium, oxidative stress, inflammation, senescence, apoptosis and alkalinization [21,22], all (except alkalinization) common manifestations of uraemia. Thus, non-surprisingly, uremic serum accelerates VC in VSMC in vitro [23,24,25].

## 3. Uremic Toxins and Their Effects on VSMC

Uraemia (in Greek) literally means “urine in the blood”. A key future of the uraemic syndrome is the accumulation of URMs. URMs comprise an extremely heterogenous group of molecules of different origins, molecular weight (MW), biological functions and physico-chemical properties [26,27]. Their classification into middle molecules (MM, MW > 500 Da), protein bound uremic retention molecules (PBURM) and low molecular weight solutes (LMWS, MW < 500 Da) has been widely accepted [28]. 

The present literature review aims to update current evidence of the role of URMs in VC [29]. A total of 151 URMs were included of which 46 were classified as MM, 32 as PBURM, and 73 as LMWS (Table 1). For only 17.2% of these URMs, the impact on VC has been investigated in an in vitro setting. In vivo and clinical data are even more limited (Figure 1). Thus, huge research gaps persist on whether and how URMs influence VC development or modify its progression 

MM are predominantly proteins and peptides generated endogenously as a response to uraemia, such as cytokines and peptide hormones. Dialysis has limited efficacy in controlling MM, partly related to size restriction of the pores of dialyzers [26]. Of the 46 MM included in this review, 9 (19.6%) have been investigated with respect to effects on VC. Of these 46, all belong to the subgroups of inflammatory cytokines and peptide hormones. A summary of the effects of MM on VSMC calcification is given in Table 2 (Appendix A). 

PBURM are a heterogenous group of solutes, characterized by reversible protein binding capacity. PBURM are considered a major threat to ESRD patients, partly because of their limited removal by conventional dialysis [30]. Of the 32 PBURM included in this review 5 (15.6) have been investigated with respect to effects on VC. A summary of their effects on VSMC calcification is given in Table 3 (Appendix A).

The group of LMWS comprises compounds with molecular weight <500 kDa and minimal protein binding. Reduction rates of LMWS during dialysis are overall high, but show substantial variability [26]. Twelve out of the 73 LMWS (16.4%) have been investigated with respect to potential effects on VC (Figure 1, Table 4 and Appendix A). LMWS have been neglected for a long time but interest in this class of URMS has increased in recent years, as it has become clear that they considerably affect patient well-being and may contribute substantially to vascular pathobiology [26]. Thus, LMWS might hold significant potential as therapeutic target, also considering that it is the largest and most understudied class with respect to VC. URM of the class MM as well as PBURM are approximately 2.5 times more often investigated compared to LMWS. That might partially be due to the fact that MM contain many non-uraemia specific molecules, e.g., inflammatory cytokines and peptide hormones also relevant for many other diseases [28]. This leads to a seemingly higher level of clinical evidence for these molecules, while the level of evidence generally remains low.

**Table 2 toxins-12-00624-t002:** Repository of studies investigating MM towards their effect on VC, as well as their effects on VC related processes in VSMC.

	Calcification -In Vitro	Calcification -In Vivo	Calcification -Clinical Studies	Osteogenesis	Oxidative Stress	Inflammation	Apoptosis	Senescence	Proliferation	Migration	Atherosclerosis
APN	↓[31,32,33] —[34]	↓[35,36]	↑[37,38,39] ↓[40,41] —[42,43,44]	↓[31,36]		↓[45]	↓[33,35]		↓[46,47]	↓[48,49]	↓[49,50]
ADM	↓[51,52]	↓[51,53]		↓[53]	↓[54,55]				↑[56,57] ↓[58,59]	↓[60,61]	
ET	↑[62]	↑[62,63,64]	↑[65,66,67]		↑[68,69,70]	↑[69,71]	↑[72] ↓[73]		↑[74,75]	↑[76,77]	↑[78,79]
IL-8	↑[80]		↑[81] —[82]			↓[83]			↑[84,85]	↑[84,85]	
IL-18	↑[86,87]		↑[88,89,90]	↑[86,87]	↑[91]	↑[92,93]			↑[94,95]	↑[96,97]	↑[98,99]
IL-1β	↑[100,102] ~[103]	↑[101,104,105]	—[101,102,106]	↑[100,102] ~[107]	—[108]	↑[109,110]	↑[111,112]	↑[102]	↑[113,114]	↑[114,115]	↑[101,116]
IL-6	↑[117,118,119,120,121,122]	↑[118,119,123]	↑[41,106,124,125,126,127,128,129]	↑[118,120,122,130]	↑[131]	↑↓[130]	↑[132,133]	↑[121]	↑[134,135]	↑[135,136]	↑[118,120]
PTH	↑[137,138]	↑[137,139,140] ↓[141,142]	↑[129,143] —[144]	↑[137]	↓[141]				—[145]		
TNF	↑[107,146,147,148,149,150,151,152]	↑[149]	↑[126,129] —[44,128]	↑[107,146,149,153]	↑[154,155]	↑[156,157]	↑[158,159]		↑[160,161]	↑[160,162]	↑[163,164]

↑ = increase, ↓ = decrease, — = tested, but no effect, ~ = unclear effect, Abbreviations: Adrenomedullin = ADM, Endothelin = ET, Interleukin = IL, Parathyroid hormone = PTH, Tumor necrosis factor alpha = TNF-α, Adiponectin = APN.

**Table 3 toxins-12-00624-t003:** Repository of studies investigating PBURM towards their effect on VC, as well as their effects on VC related processes in VSMC.

	Calcification -In Vitro	Calcification -In Vivo	Calcification -Clinical Studies	Osteogenesis	Oxidative Stress	Inflammation	Apoptosis	Senescence	Proliferation	Migration	Atherosclerosis
Hcy	↑[165,166,167,168,169]	↑[165,168,169,170,171]	↑[170,172,173,174,175,176,177] —[106,128,178,179,180,181]	↑[165,166,168,169]	↑[182,183]	↑[184,185]	↑[186,187] —[188]		↑[189,190]	↑[191,192]	↑[193,194]
IS	↑[29,80,195,196,197]	↑[195,197,198]	↑[199,200] —[201]	↑[80,196,197,202]	↑[202,203]	↑[204,205]	↑[29]	↑[29]	↑[206,207]	↑[208,209]	↑[29,210]
Leptin	↑[211,212,213]	↑[211,214,215]	↑[40,216,217,218,219] —[41,44]	↑[211,213,214,220]	↑[221,222]	↑[221,223]	↑[222] ↓[224]		↑[223,225] ↓[226]	↑[227,228]	↑[212,229]
CML	↑[230]	↑[230]	↑[231]	↑[230]		↑[232]	↓[233]		↑[233]		↑[230]
pCS		↑[198]	↑[234]	↑[235]	↑[235,236]	↑[198,235]			↑[237]	↑[237]	↑[237,238]
SM	↓[239]				↑[240]		↑[241,242]		↑[243,244] ↓[245]		

↑= increase, ↓ = decrease, — = tested, but no effect. Abbreviations: Homocystein = Hcy, Indoxyl sulfate = IS, N(6)-Carboxymethyllysine = CML, p cresyl sulfate = pCS, Spermine = SM.

**Table 4 toxins-12-00624-t004:** Repository of studies investigating LMWS towards their effect on VC, as well as their effects on VC related processes in VSMC.

	Calcification -In Vitro	Calcification -In Vivo	Calcification -Clinical Studies	Osteogenesis	Oxidative Stress	Inflammation	Apoptosis	Senescence	Proliferation	Migration	Atherosclerosis
ADMA	↓[246]		↑[247,248,249,250,251] —[252]		↑[253]	↑[254]	↑[253]		↑[254,255] —[246]	↓[256,257]	
G	—[246]								—[246]		
GAA	—[246]								—[246]		
GSA	↓[246]								—[246]		
MG	—[246]								—[246]		
MMA	↓[258]			↓[258]			↑[259]				
NA	↑[260]	↑[260]		↑[260]					↑[261,262]	↑[263,264]	
SDMA	↓[246]								—[246]		
TMAO	↑[265]	↑[265]	—[266]	↑[265]		↑[265,267]					↑[267]
UA	↑[268,269]	↑[269]	↑[270,271,272,273] —[274]	↑[268,269]	↑[275,276,277,278]	↑[276,279]			↑[280,281]	↑[282]	↑[276]
GPA	—[246]								—[246]		
GBA	—[246]								—[246]		

↑ = increase, ↓ = decrease, — = tested, but no effect. Abbrevations: γ-guanidinobutyric Acid = GBA, β-Guanidinopropionic Acid = GPA, Uric acid = UA, Trimethylamine-N-oxide = TMAO, Symmetric Dimethylarginine = SDMA, Asymmetric Dimethylarginine = ADMA, Noradrenalin = NA, Monomethylamine = MMA, Methylguanidine = MG, Guanidino succinic acid = GSA, Guanidino acetic acid = GAA, Guanidine = G.

Overall, information on the impact of uremic toxins on the process of vascular calcification, in all its aspects, is fragmentary (Figure 1) and by large incomplete; e.g., for none of the URM, information was available on the impact of either release or composition of extracellular vesicles released by VSMC, which recently gained attention as modifiers of the VC process [283]. 

## 4. Challenges in Uremic Toxin–VC Research

Experimental heterogeneity hampers comparison of data. Primary human, bovine, mouse and rat VSMCs as well as cell lines have been used by different investigators. Interspecies differences may account at least partly for inconsistent findings. VSMC derived from different topographic locations of the vascular tree may differ in their physiology and pathophysiology [284]. In addition, experimental conditions diverge substantially and may introduce substantial heterogeneity with major impact on the outcome. As an example, URM are tested as standalone or in combination with either Ca^2+^ or Pi. Also, for Ca^2+^ or Pi, concentration, donor (e.g., β-Glycerophosphate vs. Na_2_HPO_4_), exposure time and the use of foetal bovine serum (FBS) are subject to differences. A proposal for standardization has recently been put forward, however consensus has not been reached yet [285]. Additionally, the concentration of the URMs to which the VSMC are exposed matters and should ideally reflect/mimic the clinical situation. The European uremic toxin work group (EUTox) published recommendations on handling and use of URM which serves as guideline. In this respect, especially the protein bound uremic toxins deserve attention since guidelines recommend to add human serum albumin at the average uremic concentration of 35 g/L to any test system not containing protein [286]. This recommendation assumes that only the free fraction of uremic toxins exerts an effect. However, none of the publications included in this review was performed in a protein free system, but used FBS in varying concentrations containing an untested concentration of protein. Using FBS free conditions in a calcification experiment remains challenging but might be desirable to overcome this limitation. FBS also contains unknown substances and factors known to influence calcification, like Fetuin-A [287,288]. In lieu of FBS free conditions, reporting of medium protein concentrations could be an alternative and to assess the effect of the free URM fraction and foster reproducibility.

As discussed earlier, URM may affect the process of calcification also by indirectly targeting VSMC. For instance, urea and thiocyanate serve as substrates for carbamylation of LDL and proteins [289,290], an event that has recently been identified as a key driver of VC in CKD [291]. Furthermore, the multifaceted effects of especially cytokines and hormones might lead to effects relevant for VC which cannot be assessed by looking at VSMC alone, e.g., PTH (Table 2). The pleiotropic endocrine nature of hormones may not allow a credible conclusion by examining VSMC alone. Therefore, even though a direct impact of certain URM on VSMC calcification has not been established, it might not render them obsolete. Reviewing the indirect effects of URM on VSMC calcification was beyond the scope of this review. However, the above underlines the need for more research in advanced models, comprised of in vitro co-cultures and in vivo experiments. 

Literature also inheres a considerable inconsistency in effects of URM on VC between experimental and clinical studies. Biomarker assays only reflect a snapshot in time and neglect the slow progressive nature of CKD [292] and its metabolic complications, including VC. Furthermore, the relevance of systemic URM levels for the vascular microenvironment remains a concern. 

The conflicting results obtained in clinical and in vitro studies preclude strong conclusions. Furthermore, consistency of the findings is often limited. Indeed, 15 out of 26 URM have been tested only once in vitro and thus lack independent confirmation. Additionally, we found inconsistencies between clinical studies investigating the same URM. Both case-mix and variable residual confounding may account for these inconsistencies. Furthermore, clinical studies can only indicate correlation and not prove causation. In this context also correction for confounding factors might be challenging giving the multifaceted nature of CKD and the huge body of yet explored influences of the URM. Moreover, the Bradford Hill criteria [293] have not been fulfilled by any of the solutes, which underscores that current evidence is insufficient to establish a reliable cause-effect relationship, between any single URM and VC. Thus, future recommendations should be based on in vitro and in vivo preclinical and clinical research to provide impactful decisions for the advancement of quality of life for CRS patients.

Uremic serum and Pi induces phenotypic modulation and calcification of VSMC differently, thereby underscoring the importance of the uremic toxins as contributors to VC [23]. However the complexity of the composition of uremic serum and the fact that 26 different molecules have been linked to VC already, with many more having yet unclear impact (Figure 1), make it unlikely that one single substance can be identified as the “holy grail”. It is more probable that the effects leading to increased VC are attributable to an orchestra of factors questioning the effects found by testing individual molecules as being representative for the situation in situ (uremic toxin storm). It is more likely that different toxins have synergistic effects, however, might also partially antagonize each other or prime the vasculature for other toxins to be effective, without exerting an effect on its own. However, these possible ties between uremic toxins remain yet unstudied.

These findings call for a more standardized and systematic approach in order to ensure relevance of results for the therapeutic setting and to facilitate comparability between studies.

## 5. Treatment Strategies

Although there is currently no approved therapy to stop, attenuate or regress VC there are several appealing approaches [294,295]. These may be categorized as: (1) direct pharmacological inhibition of hydroxyapatite (HA) growth, (2) increasing level or activity of anti-calcifying factors, (3) reducing pro-calcifying factors. Multiple compounds for direct pharmacological inhibition of HA crystal growth are currently in development e.g., sodium thiosulfate, myo-inositol hexaphosphate or bisphosphonates. SNF472, a myo-inositol hexaphosphate formulation, was recently reported to significantly reduce progression of VC in HD-patients in a phase 2b trial [296]. Supplementation with vitamin K is under investigation to increase activity of anti-calcifying vitamin K dependent proteins. Vitamin K is an unequivocal cofactor for the gamma-glutamylcarboxylation of protein bound glutamate residues. This carboxylation is pivotal to activate the endogenous VC inhibitor matrix Gla-protein (MGP) [297]. Several clinical trials are currently underway to examine the potential of vitamin K supplementation to hold or reduce VC [294]. Additionally, also certain URM can be protective against VC (Table 2, Table 3 and Table 4), yet this has not been studied in detail. The newly emerging field of senolytics as medication might also be a promising new treatment strategy for reducing VC [298]. Renal replacement therapy as a strategy to reduce pro-calcifying factors has its limitations; e.g., the removal of PBURM by conventional haemodialysis is poor. As many vasotoxic compounds originate from gut microbial metabolism, the large intestine is increasingly recognized as an promising target of therapy.

## 6. Future Perspectives

The composition of the uremic serum differs vastly among CDK patients. Thus, although CKD patients suffer from VC, the underlying cause for VC development may differ. In order to progress towards more personalized CKD and CRS treatment strategies a better understanding of the individual uremic toxin profile is needed. This might pave the way for the early identification of patients with a higher risk for VC development and accelerated progression. Thus far, comprehensive URM profiles of individual patients are not available and information stem from measurements of a limited number of URM. Moreover, since the list of URM is constantly updated, it is difficult to define all molecules relevant in the context of VC [26]. A comprehensive understanding of the effect of individual URM is needed to identify and reduce the burden of pro-calcifying factors caused by the entire pool of URM. Since testing each URM in depth is nearly impossible, evaluating URM by classifying and clustering them to determine potential relevance will pave the way for targeted treatment strategies. We strongly put forward a structured, systematic and unbiased approach in assessing effects of URM on VSMC being most effective in revealing detrimental or beneficial effects on VC (Figure 2). Clustering of known URM based on their chemical similarity combined with proven effects might be an attractive possibility. As such, systematic screening by testing representative URM of each cluster will provide knowledge on similar URM in that cluster. Moreover, it could be worthwhile identifying common signaling pathways activated by multiple URM. This would open new avenues for counteracting the effects of URM, with very different origins and challenges for their removal. Alternatively, segregation of the overwhelming uremic toxin “storm” into manageable units might present an idea worth exploring. Such units may include the CKD-mineral bone disorder or rebalancing the inflammatory profile. A newly emerging link between intestinal microbiota and VC could also present an accessible unit [299]. A considerable fraction of the uremic toxins have been found to be gut derived [26] some of which have already been linked to VC including TMAO, IS and pCS (Table 3 and Table 4). TMAO has further been found to be indicative of intestinal dysbiosis [300], further supporting the idea of a intestine–vascular axis in CRS pathology. Therefore, a more detailed understanding of the origin of each URM might identify novel intervention sites. Moreover, senescence as feature of EVA, which has been linked to VC, should receive more attention during the design of future experiments. Extracellular vesicles are recognized as contributor to VC development, and increased vesicle secretion is associated with VSMC phenotypic switching [301,302]. Furthermore, circulating extracellular vesicles isolated from CDK patients increase VSMC osteogenic-phenotypic switching and VC [303]. Therefore, we strongly advocate for acknowledging extracellular vesicle release as process indirectly linked VC, which should receive attention also in CRS centred research. 

## 7. Summary

In this review, we included literature investigating URM and VSMC calcification in vitro, in vivo and on a clinical level. A clear definition of URM being detrimental, harmless or beneficial with respect to vascular calcification is currently lacking. Furthermore, we identify many gaps in URM research related to VSMC. A large fraction of URM has not been investigated with respect to their effects on VSMC calcification. However, some are linked to processes indirectly associated with VSMC calcification, such as ROS and inflammation. We also include potential novel avenues of VSMC driven calcification, such as extracellular vesicles, senescence and the influence of the microbiome. Effects of URM are often reported once in publications, and experimental conditions often lack an adequate degree of standardization and are not designed in relation to CRS. The composition of uremic serum is divers and may differ vastly between patients. Moreover, the influence of many uremic retention molecules on VSMC and VC, remain largely obscure and the underlying effects poorly understood. Better understanding of the effects of URM on the vasculature could be the first step towards reducing VC and a personalized CKD treatment strategy. This underscores the need for deep patient serum profiling to understand the underlying cause and consequence relationship between URM and clinical manifestations. In conclusion, our narrative overview puts forward the hypothesis to test effects of URM on VSMC in relation to VC more systematically and advocates for more CRS centred research. 

## 8. Search Strategy

The European uremic toxin work group has created an extensive database [304] containing many relevant uremic retention molecules, which served as the basis for the literature research and has been supplemented by compounds from two further publications [28,305]. The search was performed in the Medline library accessed via Pubmed using the search term “-molecule- AND (smooth muscle cell OR vascular smooth muscle cell OR VSMC OR SMC) AND (calcif* OR inflam* OR oxidative stress OR vesicle OR proliferation OR migration OR apoptosis OR necrosis OR senescence OR osteo* OR chondro* OR monocyte OR macrophage OR athero*).” If no article could be found with this strategy, at least one alternative name for this molecule has been tried in the same search term. If more than 160 articles have been found for a single molecule, the articles were sorted by “best match”, and the first 160 hits, as well as reviews from the past 10 years were included. Furthermore if publications concerning in vitro and/or in vivo results on calcification could be found, the search was extended for only these molecules to identify clinical cohort studies that relate the respective molecule to vascular calcification in a clinical setting by using the search term “-molecule- AND calcification AND cohort” as well as “-molecule- AND calcification”.

## Figures and Tables

**Figure 1 toxins-12-00624-f001:**
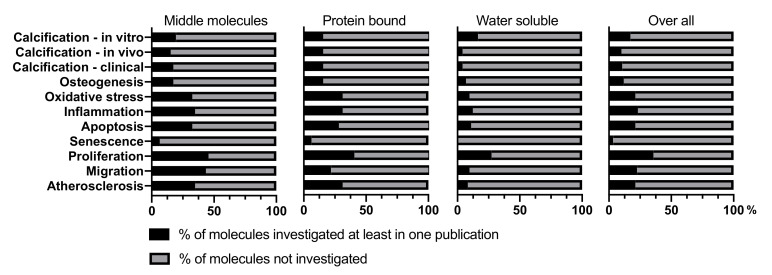
Literature was screened for effects of uremic retention molecules on vascular smooth muscle cells, with relevance for vascular calcification. Relevant review terms have been defined as the three levels of calcification research–in vitro, in vivo and clinical, and were completed by pertinent phrases. The extent to which molecules of each group have (black bar) or have not (grey bar) been investigated with respect to a given topic, displayed as % per group, is presented.

**Figure 2 toxins-12-00624-f002:**
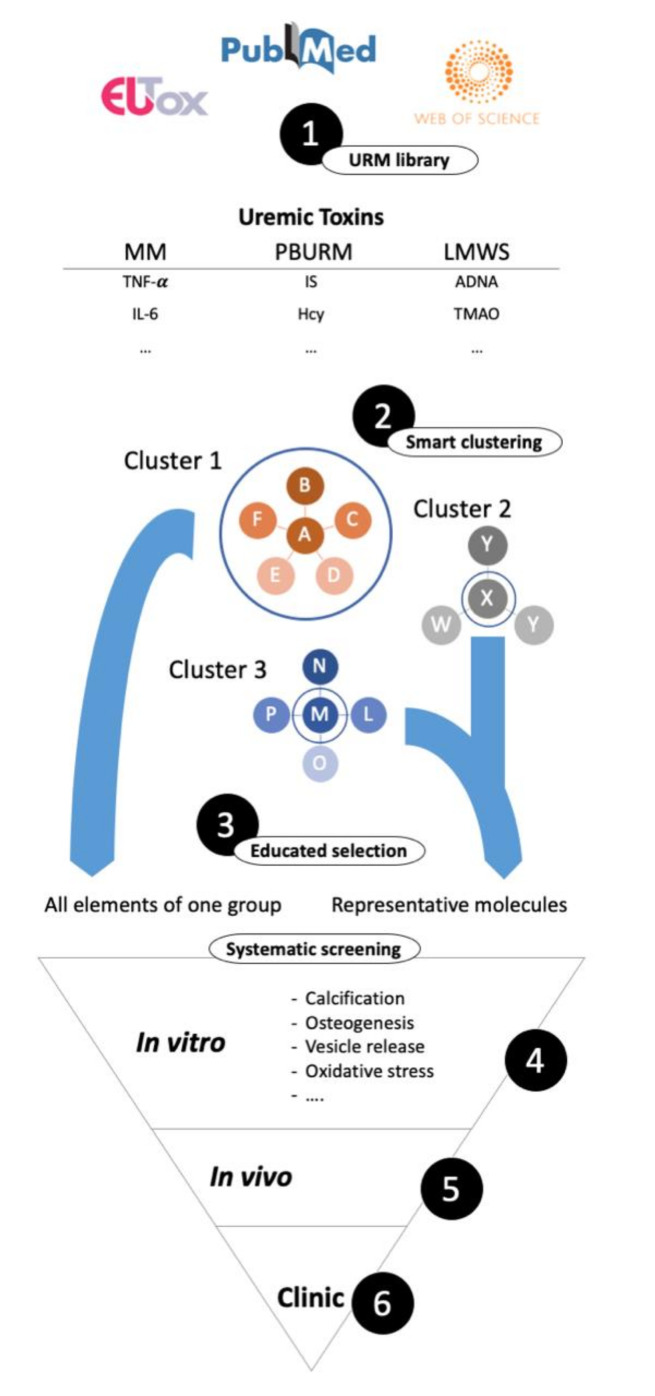
Outline of a systematic screening approach for uremic retention molecules (URM). 1. URM need to be collected and organized in a library, possibly starting with the European Uremic Toxin Work Group’s database, extended by recent discoveries. 2. URM need to be logically sorted into smaller, accessible groups. Clustering strategies could be based on chemical similarity, origin, or others. 3. A selection for actual testing needs to be applied which could be by, educated guessing based on known effects, testing the representative molecule from each cluster to identify relevant subgroups or by testing cluster wise. 4. Selected molecules are tested systematically in on one or more cell types for relevant effects like calcification, inflammation, apoptosis, etc. … in vitro, 5. If successful in vivo and 6. On a clinical level.

**Table 1 toxins-12-00624-t001:** Described URM where selected to be part of this review on basis of the database maintained by the European Uremic Toxin work group, complemented by two further publications. Table 1 summarizes the basic information of the included URM, showing the number of molecules per group (#) and the respective percentage from all molecules (%), as well as the characteristic size range and a typical example molecule.

Middle molecules	46 (30,5)	# (%)	MW > 500 Da	e.g., TNF
Protein bound	32 (21,2	# (%)	-	e.g., Indoxyl sulfate
Water soluble	73 (48,3)	# (%)	MW < 500 Da	e.g., Urea
Total	151 (100)	#(%)

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
