# Peer review of "Uremic Toxins and Vascular Calcification–Missing the Forest for All the Trees"

_toxins, 2020, doi:10.3390/toxins12100624_

Round 1
Reviewer 1 Report
The present manuscript reviews the involvement of uremic toxin on vascular calcification with potential role for early diagnosis, risk stratification and treatment of patients. Unfortunately, there is a substantial lack of standardization and vast experimental heterogeneity that hampers comparison of data.
The authors propose 1) to cluster uremic retention molecules (URM) base on similar properties as global assessment method to identify and reduce the burden of pro-calcifying factors caused by the entire pool of URM 2) to identifying common signaling pathways activated by multiple uremic toxins to study the effect of the uremic storm as unit 3) to analyze the role of intestinal microbiota on the production of uremic toxins as adjunctive therapeutic target.
The manuscript is overall well written, and the methodology applied is robust. However, I have a few minor comments that I would like to be addressed:
- In the introduction you mention 5 types of cardiorenal syndrome. Can you please briefly list them all?
- Please add a reference at line 184 for Bradford Hill criteria
- Please avoid or clarify abbreviation on figure descriptions
- Can you simplify the list of keywords that appears redundant?
Author Response
- In the introduction you mention 5 types of cardiorenal syndrome. Can you please briefly list them all?
- Line 30/ 31: inserted list of cardiorenal syndrome types
- Please add a reference at line 184 for Bradford Hill criteria
- Line 185: Added reference
- Please avoid or clarify abbreviation on figure descriptions
- Line 100-104: Removed abbreviations from description of Figure 1
- Line 255-263: Clarified one abbreviation (URM) which occurs multiple times in the figure description, removed one abbreviation (EuTox)
- Can you simplify the list of keywords that appears redundant?
- Line 17-19: Removed several key words
Reviewer 2 Report
General:
This review collects the published data regarding the effects of uremic retention molecules (URM) on vascular smooth muscle cells (VSMC) with relevance for early vascular ageing (EVA), vascular calcification (VC) and cardiovascular disease (CVD). The authors provide insights about the role of 151 URM in calcification and underlying mechanisms, including osteogenesis, oxidative stress, inflammation, apoptosis and senescence. Various uremic toxins play roles in pathological effects on the vascular wall, including endothelial cells (VCs) and smooth muscle cells (SMCs) during progression of CKD, are there specific pathological effects on endothelial cell and in addition to SMC?
Author Response
Various uremic toxins play roles in pathological effects on the vascular wall, including endothelial cells (VCs) and smooth muscle cells (SMCs) during progression of CKD, are there specific pathological effects on endothelial cell and in addition to SMC?
Yes, uremic toxins do have effects on the endothelium as well. These effects have recently been reviewed and published (doi: 10.3390/toxins12060412). Furthermore, there are effects known where uremic toxins affect endothelial cells, which subsequently affect smooth muscle cells (doi: 10.1016/j.csbj.2020.04.006). However, we consciously decided to not include these, and related, effects in the current review. The topic covered by this review shall be restricted to smooth muscle cells only, firstly to sharpen the profile of this publication, and secondly because we don't see the added value to the recent publications.